# A Qualitative Assessment of “Generacion Actual”: An HIV Community Mobilization Intervention Among Gay Men and Transgender Women in Lima, Peru

**DOI:** 10.3390/ijerph22111669

**Published:** 2025-11-03

**Authors:** Andres Maiorana, Susan Kegeles, Elizabeth Lugo, Wendy Hamasaki, Ximena Salazar, Carlos Cáceres

**Affiliations:** 1Center for AIDS Prevention Studies, Division of Prevention Science, Department of Medicine, University of California, 550 16th Street, 3rd Floor, San Francisco, CA 94143, USA; susan.kegeles@ucsf.edu; 2Center for Interdisciplinary Research on Sexuality, AIDS and Society, Universidad Peruana Cayetano Heredia, Avenida Armendariz 445, Miraflores, Lima 15102, Peru; elizabeth.lugo@upch.pe (E.L.); wendy.hamasaki.r@upch.pe (W.H.); ximena.salazar@upch.pe (X.S.); carlos.caceres@upch.pe (C.C.)

**Keywords:** HIV prevention interventions, gay men, transgender women, community mobilization, empowerment, Peru

## Abstract

The high HIV prevalence among men who have sex with men and transgender women (TW) in Peru calls for innovative HIV prevention strategies to modify social norms, increase social support and promote empowerment and community mobilization. This qualitative article presents the synergistic processes that generated community mobilization throughout Generación Actual (GA, Current Generation in English), an HIV prevention intervention with gay men (GM) and TW in Lima South based on Mpowerment, a U.S.-model intervention program. We conducted 24 interviews with GM and TW participants, informed by observations of GA and the perceptions of its implementing coordinators, and complemented by the number/types of GA activities. Four significant processes occurred throughout GA: (1) high participant engagement, community building and empowerment; (2) an effect on HIV prevention and treatment; (3) the integration of GM and TW and (4) GA’s community center becoming a safe space for socializing, support and information. These processes helped produce positive changes related to self-empowerment, personal agency and the participants’ health, suggesting an impact of GA on HIV prevention, stigma reduction and care engagement. Community mobilization strategies that ensure active community participation and involvement may constitute relevant aspects for an effective approach to HIV prevention for TW and GM in Peru.

## 1. Introduction

HIV in Peru has the characteristics of a concentrated epidemic [1], with a prevalence in men who have sex with men (MSM) estimated at 10% and in transgender women (TW) at 30%, while the incidence in these groups remains high at 5% [2,3]. Most HIV cases (78%) have been reported in Lima [4]. Although progress has been made in HIV prevention in recent years, there are still individual, social and structural factors that increase the vulnerability to HIV of MSM and TW in Peru [5]. Condom use among MSM and TW in Peru is inconsistent and is usually discontinued when in a relationship [6,7]. HIV-related stigma limits the willingness to be tested for HIV, negotiate condom use, discuss HIV status with sexual partners and engage with medical care in a fragmented and unfriendly health system if living with HIV [8,9]. The prevention efforts of the Peruvian Ministry of Health have mainly focused on the provision of condoms, management of sexually transmitted infections (STIs) and HIV testing and counseling provided by outreach workers, called “promoters.” At the time of our study, Pre-exposure prophylaxis (PrEP) was only available in Peru through clinical trials [10].

It is estimated that only between 25 and 30% of MSM and TW have been linked to some phase of the HIV care continuum, and only 18% have achieved an undetectable viral load [3,11]. The HIV care continuum includes five steps: testing to diagnose HIV infection; linkage to HIV care; ongoing care engagement; adherence to antiretroviral therapy (ART) and reaching viral suppression. ART is critical to achieve viral suppression (an undetectable amount of HIV in the blood) for people with HIV to stay healthy, maintain their quality of life and increase their life expectancy, in addition to preventing HIV transmission to others, known as “treatment as prevention” (TasP) [12].

HIV prevention and linkage to medical care strategies in Peru have not focused on interventions to modify social norms, increase social support and encourage community mobilization [13]. Community mobilization seeks to create social change through individual and collective empowerment for the members of a community to take charge of their own health [14]. Community mobilization strategies, designed to engage and galvanize community members to take action towards the achievement of a mutual goal, are increasingly seen as core components of combination HIV prevention programs [15].

In countries with a concentrated epidemic, community mobilization is considered an essential component to implement effective and innovative interventions that address the social context related to HIV, develop social cohesion, expand social networks, reduce stigma and ensure community participation in HIV prevention and care [16]. Community mobilization is usually successful in making visible the social vulnerability that increases the risk of acquiring HIV and in facilitating the acceptance of STI and HIV testing, condom use and treatment [17,18,19,20]. However, there is a need to better understand the processes and mechanisms of community mobilization to refine this approach [21].

### 1.1. Proyecto Orgullo+ (PO+)

The HIV prevention strategy for MSM/TW in Peru has not included broadly accepted approaches, such as the combination of biomedical, behavioral, social/community and structural interventions [22,23] and the conceptualization of HIV prevention and care as a continuum. Because of the limits of this strategy, HIV researchers at the University of California and Cayetano Heredia University in Lima designed and implemented PO+ (Project Pride) [24,25]. PO+ is a multilevel intervention whose main goal was to strengthen the continuum of HIV prevention and care in gay men (GM) and TW that was implemented in the southern area of Lima (Lima South). PO+ consisted of two innovative parts: (1) a health system component carried out at two health facilities serving the area that included an HIV peer navigation program for patients [26,27], training for health providers on the HIV care continuum and the needs and vulnerabilities of GM/TW and written materials on the importance of TasP for both patients and providers; and (2) “Generación Actual” (GA), Current Generation in English, the community mobilization component described below. This paper only focuses on GA.

PO+ was implemented in San Juan de Miraflores (SJM), a bustling commercial and residential low-income district of Lima South, with a significant number of migrants from different regions of Peru, including the highlands and jungle areas, looking for better living conditions by moving to Lima [28]. We selected SJM in Lima South to conduct this project because it is a smaller geographically defined area with a sense of community within the much larger city of Lima and accessible by public transportation. Based on ethnographic mapping, we identified that there was a sufficiently large population of GM and TW in SJM to make it feasible to implement PO+, but only nascent LGBTQ+ organizations and support networks, and few health and HIV prevention activities oriented towards the LGBTQ+ community in the area.

### 1.2. Generación Actual (GA)

GA—the name was given to the intervention by the participants themselves—was implemented for a little more than three years (July 2017–September 2020) in SJM to generate community mobilization around health, including organizing GM and TW to support each other to increase engagement in HIV prevention efforts (e.g., using condoms, testing for HIV) and linkage to care. As a precedent for PO+ and GA, we conducted formative research and implemented a nine-month pilot intervention in 2013 in the port area of Callao (Lima), which demonstrated the feasibility and acceptability of a community intervention such as GA among GM and TW [29]. While historically, GM and TW in Peru have not worked well together due to mutual assumptions and prejudice, an objective of GA was to foster collaboration between these two populations towards a common goal: community building and community mobilization related to HIV prevention and care.

The pilot in Callao and then GA were an adaptation of Mpowerment (MP), a group-level HIV prevention community intervention for gay men widely and successfully implemented in the USA and in other countries [30,31]. MP was originally designed as a primary prevention strategy focused on reducing unprotected sex, but as it is multilevel and inherently flexible, its components are well-suited for delivering health-promotion content across a variety of domains, including HIV care. The theoretical principles of GA [24,32,33,34] are described in Table 1, along with its seven core elements [24], designed to act synergistically for the intervention to function effectively in Table 2.

Since GA is a community-level intervention, all project activities were open to all GM/TW living in Lima South. Gamification approaches were used to promote active participation in GA. Three of the GA core elements, (1) the four-session experiential workshops; (2) the core group [Grupo Impulsor (GI) in Spanish], the decision-making body of GA comprising the coordinators and participants who volunteered for it and (3) the mobilization activities part of formal outreach, used gamification to make activities fun and enjoyable rather than boring or tedious discussions about HIV. These approaches included exercises, presentations and discussions conducted with positive, uplifting approaches and affirmation and pride regarding issues related to sexuality, gender, internalized homo/transphobia, sexual health and HIV-related issues. The objective was to facilitate self-reflection and critical consciousness as well as individual and community empowerment in ways that increased the interest of participants. While being enjoyable, the project also strived to weave in the development of critical consciousness into the activities, using discussions to help participants reflect further on their lives, relationships and HIV-related health behaviors. The HIV issues addressed had been previously identified as being related to HIV prevention and treatment if living with HIV.

Guided by our previous formative research and the results of the pilot in Callao, we conducted adaptation work before the start of GA to contextualize the intervention to respond to the specific needs of GM and TW in Peru. Different from the MP project, the core group, instead of being limited to a group of approximately 10 men, because of a more collectivist social ethos in Peru was open to anyone who wanted to participate and collaborate in organizing mobilization activities. Other adaptations, because of the interest expressed by Peruvian GM and TW, encompassed expanding the HIV single session from the MP project addressing HIV into four experiential workshops addressing other issues such as sexuality and gender. GA included both joint and separate activities for TW and GM. Most of the exercises, games and role plays in the experiential workshops were adapted from the pilot [29] or newly developed for GA. They were pilot tested with GM and TW. For example, the drawing exercise “Looking for Mr. Perfect” in the experiential workshops for GM promoted reflection and discussions about seeking sexual or romantic partners and communication with those partners, including negotiating condom use and disclosure of HIV status. The parallel exercise “Looking for the Perfect Body” in the workshops for TW asked participants to draw the perfect body they wanted for themselves in their physical transition. It promoted reflection on the reasons for wanting that perfect body, whether for themselves or for a male partner, and discussions about the use of silicones and hormones without medical supervision to achieve that body. This exercise was followed by a video about the risks of using industrial silicones.

The curriculum of the experiential workshops included the following objectives related to HIV: increasing HIV literacy and positive perceptions about HIV testing and treatment to achieve and maintain viral suppression; fostering sexual communication and condom negotiation skills with primary and non-primary partners; increasing disclosure of HIV status to partners and reducing internalized HIV and intersectional stigma. The importance of discussing these issues with peers in the community, in concordance with the core element of informal outreach, and how to maintain those conversations were also addressed in the workshops. Similarly to the pilot in Callao, attendance at the workshops was considered as the minimum dose of the HIV-related content part of the intervention.

The three coordinators (two GM and one TW), paid staff who had previous experience working with the gay and transgender communities, were trained on the guiding principles and core elements of GA, and how to implement the curriculum of the experiential workshops. The coordinators attended periodic booster trainings throughout the intervention. We did not provide financial incentives to participants in GA. Since Peru was one of the South American countries most affected by the COVID-19 pandemic, the activities in the last seven months of GA implementation (March–September 2020) were conducted virtually because of the strict quarantine measures implemented by the government [35]. These measures impeded the final quantitative outcome assessment of a longitudinal cohort designed to determine the impact and dissemination of GA throughout the community among TW and GM in SJM, independently from their specific participation in GA.

The purpose of this qualitative analysis is to examine four significant processes that occurred throughout the implementation of GA. These processes are as follows: (1) active participation, community building and personal and community empowerment; (2) the effect on HIV prevention and treatment; (3) the integration of gay men and transgender women and (4) the community center as a safe space generating community mobilization and positive changes in the participants. This study was approved by the Institutional Review Boards of the Universidad Peruana Cayetano Heredia and the University of California in San Francisco.

## 2. Materials and Methods

We situate this study within an interpretivist and constructivist research paradigm intrinsic to qualitative research as a framework to understand and interpret socially constructed subjective realities, as well as deriving meaning from the participants’ experiences, perspectives and social interactions [36] related to their participation in GA in the context of their lives. In this article we present qualitative data from interviews conducted with GM and TW who participated in the activities of GA, informed by observations of GA implementation and the perceptions of the GA coordinators, the paid staff who ran GA, about implementation. In addition, we present information about the number and different types of GA activities to complement the qualitative findings. All data were collected as part of a Process Evaluation informed by Moore’s model [37], conducted throughout GA, to document and evaluate its feasibility and acceptability and implementation. All authors are researchers experienced in conducting public health studies and HIV research with GM and TW and other vulnerable populations in Latin America.

### 2.1. Data Collection

We conducted in-depth interviews with a purposive sample of 24 GM and TW to examine their perceptions and experiences related to their participation in GA. Purposive sampling is applied in qualitative research to explore specific experiences in detail. The selection of information-rich participants for purposive sampling is not random but intentional and based on their knowledge, specific qualities or characteristics important to the study [38]. We asked the coordinators in charge of organizing GA to identify TW and GM of different ages participating in different waves of the experiential workshops and other GA activities to be part of the interviews. The coordinators introduced the potential participants to the interviewers. All the potential participants identified, who had not met the interviewers in advance, agreed to be interviewed. However, because the interviewers also conducted observations, the potential participants may have noticed them and known their names prior to being contacted to be interviewed. The interviewers screened the potential participants, explaining the purpose of the interview before arranging a day and time for the interview. Conscious of potential power dynamics, when first meeting the potential participants and then during the interviews, the interviewers strived to establish rapport and trust and a horizontal relationship with them. The number of participants corresponded to the initial number of interviews proposed in the design of the evaluation.

The two interviewers (E.L. and W.H.) oversaw all process evaluation related procedures and were part of the analysis team, which facilitated immersion in the data. The interviewers were not involved in implementing GA. Before data collection, conducting mock interviews with four different team members of PO+ helped to refine some of the questions in the semi-structured interview guide. The guide incorporated domains corresponding to the goals of the evaluation. The domains included participants’ perceptions and experiences in different GA activities, the influence of GA at a personal level (self-reflection and self-empowerment; and HIV prevention, testing and treatment engagement) and at a community level (mobilization, community building and joint work of GM and TW) (See participant interview guide as Appendix A). Participants provided verbal consent before the interviews, which lasted an average of 45 min and were conducted in a private space at the GA community center when no other participants were present and no activities were taking place. All interviews were conducted in Spanish and were audio-recorded and transcribed verbatim by a professional transcriber. All transcripts were de-identified. We provided a small incentive of 10 PEN (3 USD) to the participants in the interviews. The field notes written during the interviews were useful during the analysis process.

The interviews were conducted on a rolling basis between February 2018 and February 2020. This gradual approach was used to include in the sample TW and GM participants who attended the activities at different times throughout the implementation of GA. We observed approximately 50% of all GA activities, including those implemented virtually in 2020. We used an observation sheet organized by date and activity to record interactions and processes taking place during the implementation of GA to assess intervention delivery and participant engagement. Observations were unobstrusive of the participants. Additionally, we used notes from approximately 75% of the weekly supervisory meetings with the coordinators that helped capture their impressions and opinions about GA implementation and GM and TW participation in the intervention. We used a registration sheet to collect anonymous demographic information (age, sex/gender, neighborhood of residence) and to record the number of participants attending the different types of GA activities.

### 2.2. Analysis

The analysis process was iterative, using an inductive and deductive approach based on thematic analysis [39], carried out by A.M., E.L. and W.H., native Spanish speakers trained in anthropology, public health, psychology or sociology, and with experience conducting qualitative research. Thematic analysis provides a flexible, rich and detailed approach to examine the perspectives of research participants, allowing us to emphasize similarities and/or differences in the data and generating themes and thematic interrelations [40]. Consistent with the tenets of thematic analysis, the analysts first independently coded the transcripts of four interviews and developed a preliminary coding book based on the interview guide questions but also including themes emerging from the interviews. Then, they discussed their coding as a group and solved discrepancies. Using that initial coding, they developed a final codebook that was applied to the rest of the interviews coded using Dedoose, a cloud-based secure online application, useful for organizing, visualizing and analyzing qualitative data in real time as a team [41]. The analysis team met virtually and periodically to discuss coding, share impressions and interpret results. In summary, the process included the following steps to ensure rigor, consistency and cohesion in the analysis and its validity, credibility, trustworthiness and confirmability: familiarization with the data; creating a codebook; identifying, revising and defining themes; creating code reports; establishing relations among themes; tracking methodological decisions in a journal and writing up results focusing on relevant themes [42]. Keeping a journal and debriefing and self-reflection as a team contributed to reflexivity throughout the analysis process [43]. Consensus among a team of researchers from different disciplines about understanding and interpreting the same reality increases credibility, as well as ensuring that the congruence of the analysis is strong and solid [44].

Observations and notes from supervisory meetings were used to supplement interview data. Observation data were shared with the intervention supervisors to inform GA implementation (e.g., assessing accomplishment of goals and flow of activities, reminding coordinators to integrate everyone in the GA activities and welcome all new participants). We first summarized these observations and notes, organized in Word documents, to capture themes related to GA implementation and participant engagement. Triangulating these data with interview data showed similarities and consistency between the perceptions of the participants interviewed and the coordinators, as well as our observations related to community mobilization. The convergence of the information among these data sources adds validity to our findings. The participants’ demographic information and attendance at GA activities were first processed in an Excel spreadsheet and then organized in tables. In the results, we report on four themes, mainly based on analytic codes related to community mobilization, sexual health, HIV prevention and care, community building and empowerment.

## 3. Results

The interviews included 24 participants [15 GM (62.5%) and 9 TW (37.5%)] as a group that reflected the relative proportions of GM and TW participants in GA. Interview participants were between 18 and 57 years of age. The majority resided in the district of SJM or in a few of the surrounding districts. They were of a low socioeconomic level. Many worked in different areas of the informal economy, in shops or restaurants, as street vendors and hairdressers or doing sex work. They had educational levels ranging from incomplete primary school to technical school or college. Participants who identified as TW varied in their gender presentation.

In the following sections we present four main themes to characterize the significant processes related to the implementation of GA that facilitated community mobilization. These processes included (1) active participation, community building and personal and community empowerment; (2) the effect of GA on HIV prevention and treatment; (3) the integration of GM and TW in GA and (4) the GA community center as a safe socialization space in SJM. We use pseudonyms to refer to the participants we quote. Together with their English translation, we include the original quotes in Spanish to better represent the participants’ voices.

### 3.1. Active Participation, Community Building and Personal and Community Empowerment

Group exercises, games and discussions in the periodic two/three-hour experiential workshops, conducted separately for GM and TW to develop trust and rapport and facilitate self-reflection, helped participants not only to learn more about HIV and how to prevent it, but also to better understand and cope with the impact of social oppression and intersectional or compounded stigma related to HIV and their sexuality or gender expression, as well as minority stress [45] on their lives. The workshops were conducted in “waves” of approximately ten participants. A total of 210 persons (128 GM and 82 TW) participated in them (See Table 3). This also represents the minimum number of unduplicated participants in GA, since the workshops were designed, although not exclusively, as a gateway to GA participation.

An average of 21 GM and TW, together with the coordinators, participated in the weekly GI meetings. The GI met at the community center to express opinions and ideas, propose and organize new mobilization activities based on common objectives, assign functions to volunteers and evaluate previous activities. The community mobilization activities were chosen by vote and focused on topics of interest to the majority in the GI. Referring to active participation and horizontality in the decision-making, Salvador stated the following:

“What I like about GI is the joint participation, it’s not that they tell you ‘you know what, we are doing this activity’, but rather that from the newest to the oldest they give their opinion, help with the decoration, they stay to cook. There are people, so to speak, leaders [coordinators], who help the guys to better articulate, always in every group, there are leaders, who help to motivate others.” (GM, age 35)“Lo que me agrada del GI es la participación en conjunto, no es que te digan ‘sabes que, se hace esta actividad y la hacemos’, sino que desde el más nuevo hasta el más antiguo dan su opinión, ayudan en la decoración, se quedan para cocinar. Hay personas, por decir, líderes, que hacen que los chicos como que se articulen, siempre en todo grupo, que ayudan a que los demás se motiven.” (GM, 35 años) 

The GI not only decided which community mobilization activities the project would hold, but implemented them as well, along with any other volunteers who wished to collaborate. The mobilization activities were of different sizes (small, medium and large) (see Table 4), had different regularity and periodicity and included organizing large gatherings on special occasions such as Halloween and Mother’s Day. Activities directly linked to HIV involved helping participants analyze personal, interpersonal and social aspects related to vulnerability to HIV and taking community action to support friends in relation to HIV prevention and treatment.

The participants developed a greater sense of ownership of GA over time. Towards the beginning of GA, the coordinators defined the weekly agenda and facilitated the GI. As the intervention progressed, they promoted and encouraged the co-facilitation of the GI and the mobilization activities with the participants to nurture the development of skills and processes of reflection, self-management and individual and community empowerment. Throughout the intervention, the coordinators emphasized the need for the participants to take on the project and its objectives as their own by becoming more involved in it, taking responsibility for organizing and participating in activities; telling their friends about the project and inviting them to participate in it; and recognizing the importance of the integration and joint work of GM and TW to strengthen the community and improve the continuum of care as a goal of GA.

Trust in the coordinators resulted in participants confiding in them privately about personal issues, including disclosing their HIV status and requesting advice about receiving treatment. Sharing common experiences and stories in the GI meetings and mobilization activities also facilitated that participants were more engaged and involved in the GI, taking on tasks, preparing materials and organizing and facilitating community mobilization activities. Based on the perceptions of the coordinators and the participants we interviewed, as well as on our observations, we infer that active participation increased and gradually consolidated throughout GA implementation, favoring the development of assertive communication and respect among participants. The gradual development of trust both between participants and coordinators and among the participants themselves also contributed to increased participation and the development of social networks. While the level of participation and involvement in GA among many participants was high and ongoing, participation of others fluctuated. It was influenced by different factors including motivation and interest, time limitations due to studying or working and moving to other districts or cities. Participating in larger LGBTQ+ community events as members of GA, such as having a GA float at the yearly Pride parade, also helped to increase awareness about HIV and human rights and was conducive to empowerment and community building. After volunteering at a GA booth in a health fair, the two participants below explained the following:

“We discussed an issue of HIV prevention, near SJM, I participated and provided information, it was interesting. A little tiring, because putting everything together requires a lot of energy. But you leave gratified, right? Having helped.” (Diego, GM, age 33)“Tratamos un tema de prevención de VIH, cerca de SJM, yo participé e informé, fue interesante. Un poco cansado, porque para armar todo se necesita bastante energía. Pero uno se va gratificado ¿no? De haber apoyado.” (Diego, HG, 33 años)

“I have participated in the AIDS campaigns; it was very good. We gave them knowledge and we addressed them in the following way: ‘Sir, good morning, we want to give you information about condom day, today there is a distribution of condoms, and they are used like this, like that. Please take a couple and your lube,’ and they played roulette [a game about HIV awareness], and we asked them questions. If they didn’t know what to answer, we gave them the answers.” (Aurora, TW, age 50)“He participado en las campañas del SIDA; me pareció muy bien. Les dimos conocimientos y abordábamos de la siguiente manera: ‘Señor, buenos días, queremos informarle por el día del preservativo, hoy hay repartición de condones, y se usa así, asá… por favor, tenga un par y su lubricante’, y jugaban a la ruleta [juego acerca del conocimiento del VIH], y les hacíamos preguntas. Si no sabían qué responder, les dábamos las respuestas.” (Aurora, MT, 50 años)

The presentations, discussions and reflections also related to civil rights and the importance of community action to ensure those rights, such as procedures about how to address the authorities in case of violations of rights, that took place both in the GI meetings and in the mobilization activities may have contributed to strengthening GM and TW’s empowerment, personal agency and their ability to exercise their rights and defend their freedom of expression in the face of situations of discrimination or violence. This is particularly relevant for disenfranchised TW of a low socioeconomic level, who are marginalized and suffer societal transphobia and violence in the streets, including by police [46,47]. Nelly explained the following:

“Before I didn’t know about my rights, I work on the street, and I said it [in GA]: ‘this and this other thing happened to me.’ They advised me how to react to a police officer, now I have more knowledge that has helped me” (Nelly, TW, age 22)“Antes no sabía de mis derechos, yo trabajo en la calle, y lo contaba [en GA]: ‘me ha pasado esto que el otro’. Me aconsejaban cómo reaccionar ante un policía, ahora tengo más conocimientos, eso me ha servido” (Nelly, MT, 22 años)

Similarly, Juani stated that participating in GA helped her feel freer outside of GA too, with less fear of rejection and discrimination:

“I feel I have grown, because before [GA] I couldn’t go out anywhere, I was afraid, there is discrimination against the girls [TW] and I thought ‘something is going to happen to me if I walk alone, what if they kill me’, but then little while little by little, I totally grew. Now, normal, I walk free.” (TW, 24 years old)“Me he desenvuelto más, porque antes [de GA] no podía salir a ningún lado, me daba miedo, hay discriminación hacia las chicas [mujeres trans], y yo pensaba ‘me vaya a pasar algo si camino sola, qué tal me matan’, pero después poco a poco, me desenvolví totalmente. Ahorita, normal ando libre.” (MT, 24 años)

At the end of 2019, the GI took the initiative to organize a weekend retreat to decide what activities to continue as a group, empowered by the experience of participating in GA, after the end of the project in 2020. Participants in that retreat decided to take steps to develop their own organization to continue working together on health issues and community building after the conclusion of GA as a research project. To that end, the GI contacted authorities from the municipality of SJM, seeking to start making itself visible locally as a new sexual diversity organization and potentially obtain external funding. However, even if the GI had secured the use of a municipal space for its future activities, support of the municipality and other authorities in Lima South did not materialize due to other emerging governmental priorities during the COVID-19 pandemic. After the closing of GA, we know through the coordinators that some participants remained in contact with them, attending together sexual health activities organized by existing LGBTQ+ organizations. At that point in time, however, no clear lines had been determined to establish their own organization, as participants may have had other priorities, reorganizing their lives and seeking to earn an income if they had not been able to work during the COVID-19 quarantine.

Because of the COVID-19 pandemic, in March 2020 GA shifted to virtual activities; the only in-person activities consisted of food distribution to participants in need. These virtual activities at first consisted mainly of emotional support provided by the coordinators to the participants, as well as among the participants themselves, through WhatsApp to counteract the isolation and mental health issues arising during the quarantine. Then, from May until September 2020, when GA ended, some of the mobilization activities continued virtually three or four times a week, using Facebook Live, Zoom and groups on WhatsApp. GA provided some participants with technical assistance to help them access the Internet and virtual platforms. GA also provided mobile phone prepaid recharges every 15 days to participants who requested them and attended the online activities frequently.

Some of these activities, such as GI meetings, Feminine Mondays, Gay Saturdays and movie nights, as included in Table 4, while virtual, continued as previously in person at the community center. The topics of other activities were new, such as workshops about intersectional stigma related to HIV and COVID-19, and presentations and discussions on leadership and community organizing in response to the GI’s intention to create its own organization. Based on our observations, although perhaps influenced by the need for contact and communication with other people during lockdown, engagement in the virtual activities was high, not only of previous participants but, as a sign of diffusion of GA in the community, also of other GM and TW in Lima South who had not been involved with GA earlier.

In April 2020, GA distributed food baskets to many GA participants who were experiencing food insufficiency. Then, from June to August 2020, a coordinator and some participants took the initiative to organize a food kitchen funded by PO+ and donations from community organizations and civil society. The shopping, cooking and distribution of food to approximately 60 GM and TW three times a week, maintaining appropriate sanitary and safety precautions, were carried out by GA participants and a coordinator. The coordinators reported how grateful and appreciative food recipients were of the aid, knowing that GA and their peers in the community had thought of them and their needs. These activities during the pandemic reflected not only resilience and solidarity but represented a mobilization effort resulting from GA to maintain and strengthen the community, supporting GM and TW who had very low resources, and who for a variety of reasons had not received the bonuses provided by the government during the quarantine.

### 3.2. Effect of GA on HIV Prevention and Treatment

Presentations, discussions and messages related to HIV prevention and treatment were embedded in most of the GA activities. Only a few participants knew about the existence of PrEP when it was mentioned as a prevention strategy during GA. Similarly, participants were not familiar with the concept of undetectable equals untransmittable (U=U) [48] and the importance of treatment to reach viral suppression if they were living with HIV. GM and TW stated that, because of their participation in GA, they experienced changes on a personal level regarding HIV prevention and sexual health in general, as discussed in presentations such as “Taking Care of Your Behind”. Aurora, a sex worker, referred to using condoms with every sexual partner because of what she had learned in GA:

“…Before I was like a lost sheep, not now... they have taught me [in GA] to know how to take care of myself, [so] with any guy I use condoms.” (TW, age 50)“… Antes era como una oveja descarriada, ahora no… me han enseñado [en GA] a saber cuidarme, con cualquier chico uso preservativo.” (MT, 50 años)

Astrid, on the other hand, referring to herself in the third person, as some people in Peru express themselves, stressed the importance of taking HIV treatment, as a message she had received in GA:

“… If girls [TW] have HIV, they should not stop taking treatment so that they can be well. “They give us those messages in the community house [ referring to GA].” (TW, age 24)“… Si las chicas [mujeres trans] tienen VIH, no deben dejar de tomar el tratamiento para que puedan estar bien. Esos mensajes nos dan en la casa comunitaria [refiriéndose a GA].” MT, 24 años)

GA served as a space for emotional containment and support related to HIV. The conversations, reflections and messages around HIV contributed to talking more openly about a healthy sexuality, gradually understanding social and internalized HIV stigma, normalizing living with HIV and revealing one’s serological status. The fear of openly discussing HIV began to subside as relationships of trust were generated and issues related to HIV stigma and the fear of being stigmatized because of having a HIV diagnosis were addressed. As some participants expressed, the understanding and friendship bonds forged within the community center, which was presented as a place free of stigma, contributed to reducing the negative perception of an HIV diagnosis and living with HIV. Consequently, some participants disclosed their HIV diagnosis openly within GA.

“Coming [to GA] helped me a lot when I found out about my diagnosis, because here they lift your spirits a lot. I can come and have people with whom I can hang out and talk about my things…” (Elvis, GM, age 26)“Venir [a GA] me ayudó mucho cuando me enteré de mi diagnóstico, porque aquí te levantan mucho el ánimo. Puedo llegar y tener personas con las que me puedo juntar y conversar de mis cosas…” (Elvis, HG, 26 años)

“... I am no longer afraid, like at the beginning... to reveal something personal... I am a person who lives with “the condition” [HIV] and as I tell you, it does not limit me to anything, and [Participation in GA] has helped me to believe in me and I am studying my second degree.” (Matías, GM, age 38)“... Ya no me da miedo, como al comienzo… revelar algo personal… soy una persona que vive con “la condición” [VIH] y como te digo no me limita a nada, y me han ayudado [Participación en GA] a creer en mí y estoy estudiando mi segunda carrera.” (Matías, HG, 38 años)

In relation to the GA core element of informal outreach, and based on participant testimonies and our observations, we know that participants invited their friends to GA and had conversations with their peers about HIV. Some participants reported that the learning gained at GA helped them see the importance of talking about safer sexual practices, HIV testing and treatment with their peers by providing them with information in their interactions outside of the GA community space.

“I want to continue learning [about sexual health] to help my community, and the girls who come after me, 15 to 20 years old, who do not have any guidance. I would have liked this to happen [referring to GA] when I was 17 years old, and I didn’t know what to do or what to say. I think that talking to my friends about transphobia and HIV is to encourage them to also want to participate [in GA]” (Evelyn, TW, age 57)“Quiero seguir aprendiendo [sobre salud sexual] para ayudar a mi comunidad, y a las chicas que vienen detrás de mí, de 15 a 20 años, que no tienen ninguna orientación. Me hubiera gustado que esto ocurriera [refiriéndose a GA] cuando tenía 17 años y no sabía ni qué hacer ni qué decir. Creo que hablar con mis amigas sobre transfobia y VIH es incentivar a que ellas también quieran participar [en GA]” (MT, 57 años)

“…I have participated in all the activities [at GA]. We talk about health and we focus on HIV. It is an important issue because we are a vulnerable population. I, as a person with HIV, by being informed I can help others.” (Matías, GM, age 38)“… He participado en todas las actividades [en GA]. Hablamos sobre salud y nos enfocamos en el VIH. Es un tema importante porque somos una población vulnerable. Yo, como persona con VIH, al estar informado puedo ayudar a otros.” (HG, 38 años)

### 3.3. Integration Between Gay Men and Trans Women

GA facilitated GM and TW working together towards community building and community mobilization related to HIV prevention and care. This integration process was a gradual result of coexistence and sharing, allowing both groups to get to know each other better, actively participating in the GI and mobilization activities and reflecting on homophobia, transphobia, discrimination and the importance of cooperative work. It was achieved because of the coordinators as role models, together with the close and regular contact between GM and TW in the community center, which helped to distance them from their mutual prejudices. As part of GA, separate activities took place one day a week for TW (Feminine Mondays) and another day for GM (Gay Saturdays); while the other days, GI meetings and mobilization activities were jointly held together. At the beginning, several GM and TW openly expressed that they did not feel comfortable attending joint activities. For some GM, it was the first time they met and interacted with TW, and they had to overcome transphobia, which decreased as they got to know TW also participating in GA. Some TW also indicated that they had to overcome their distrust of GM and their fear of being judged or belittled. As time went by, coexistence, socialization and collaborative work improved communication and camaraderie between both groups. Both GM and TW acknowledged that those prejudices reflecting internalized homophobia and/or transphobia prior to their participation in GA had decreased.

“Before coming here [GA], I was like, excuse me, I was disgusted by trans women, I was afraid, I didn’t want to be by their side, I said... ‘why would you do that [sex work], how disgusting!, why do you think you’re a woman if you’re not woman’, thus ignorantly. Here I met several trans girls who I love very much now. My thinking totally changed, now they are my friends.” (Efraín, GM, age 19)“Antes de venir acá [GA], tenía como, discúlpame, tenía asco a las trans, tenía miedo, no quería estar a su lado, yo decía… ‘por qué trabajará en eso, ¡qué asco!, por qué se cree mujer si no es mujer’, así ignorantemente. Acá conocí a varias chicas trans que ahora las quiero mucho… mi pensamiento cambió totalmente, ahora son mis amigas.” (Efraín, HG, 19 años)

“We were more distant, if we greeted each other, it was a ‘hello and bye’, they told me that gay boys [GM] were reluctant to have friendships with trans women, something that I have not seen now [in GA], by telling you that I have gay friends, I love them and we have formed a good camaraderie, sometimes we meet at their homes. If something has grown, it is precisely that, the community, the communication. (Evelyn, TW, age 57)“Éramos más distanciados, si nos saludábamos era un ‘hola y chau’, a mí me decían que los chicos gays eran reacios a tener una amistad con las mujeres trans, cosa que ahora no he visto [en GA], con decirle que tengo amigos gays, que los quiero y hemos hecho un buen compañerismo, a veces nos reunimos en sus casas. Si algo ha crecido es precisamente eso, la comunidad, la comunicación. (Evelyn, MT, 57 años)

This integration and mutual support made it possible for GM to empathize with aspects that, for TW, carried a particular meaning. A specific example was the organization of “quinceañeras” (a celebration for girls turning 15) for a couple of TW, who while older, expressed the wish to have the party they could not have when they turned 15 years old, as a ritual of passage that carried a meaning of validation and social acceptance of their gender expression. In those occasions, a few GM took the initiative and committed to successfully organize those events. Darío, one of the organizers explained the following:

“…There are things that seem insignificant, but for one it is [fulfilling] an illusion that they can make it happen for you. See the world from another perspective. For example, on Saturday we will have a quinceañero, which came out of a meeting between the guys and the girls. A girl said, ‘my birthday is on such a day’ and then we said, ‘why don’t we make your quinceañera?’, and another guy added ‘hey, I’ll give you the dress’; ‘I’m going to be your godmother’; ‘I’ll give you the cake’; ‘I’ll give you the bouquet’. So everyone collaborated and the party was organized…” (Darío, GM, age 38)“… Hay cosas que parecen insignificantes, pero para uno es una ilusión que te lo puedan realizar. Ver el mundo desde otra perspectiva. Por ejemplo, el sábado se hará un quinceañero, y salió de una reunión entre chicos y chicas. Una chica dijo, ‘mi cumpleaños es tal día’ y ahí dijimos, ‘¿por qué no hacemos tu quinceañero?’, y otro chico agregó ‘oye yo te doy el vestido’; ‘yo voy a ser tu madrina’; ‘yo te doy la torta’; ‘yo te doy el bouquet’. Y así todos colaboraron y se armó la fiesta…” (Darío, HG, 38 años)

### 3.4. The Community Center as a Safe Space

The project rented a space in a central area of SJM to function as a community center, where most GA activities were held (GI meetings, movie nights, etc.), except some, such as outings to the beach or the theater and participation in LGBTQ+ community events. This social space, as one of the core elements of GA, allowed GM and TW to meet, receive information about sexual health and HIV, obtain condoms, socialize, exchange personal experiences and share moments of reflection. The center was equipped with a kitchen, computers with Internet access and a film projection system. The walls were painted and decorated by some of the participants, capturing drawings that symbolized the appreciation of and support for sexual and gender diversity and empowerment.

The functional dynamics of the center also made it possible for the participants to gradually identify with the project, taking ownership of the place and coming to consider it as their “second home”, a social space where they could be “themselves” safely, without fear of being singled out or judged for their sexual or gender identity. One participant remarked the following:
“I feel like I am coming home. I have changed a lot, I was quieter, I was afraid, ashamed and now, no. You see another Paola, a new Paola.” TW, age 26“Siento como que estoy llegando a mi casa (…) He cambiado bastante, yo era más callada, tenía temor, vergüenza y ahora, no. Ves otra Paola, una nueva Paola.” (MT, 26 años)

Likewise, participants developed a sense of belonging and a symbolic appropriation of the space, when, for example, they collected funds to buy a stove for the community center and some of them began to take responsibility for the preparation of the dinners distributed at the end of the mobilization activities. Cooking classes were also organized by some participants, and the dishes prepared were shared among the attendees, providing moments of learning, skill development (perhaps to cook creative but unexpensive dishes at home) and socialization and integration between GM and TW. As reported by one of the participants in charge of the cooking, the meals also covered basic needs in a safe place away from potentially risky environments:
“…the help that the community center provides is the dinners. When we met, I offered myself and told [name of coordinator] ‘buy rice, meat, and I’ll prepare rice with meat.’ There were guys and girls who do not have [resources], and instead of being on the street, it is better that they are here, learning.” (Darío, GM, 38 years old)“… la ayuda que brinda la casa comunitaria son las cenas. Cuando nos reuníamos, me ofrecía y le decía a [nombre de coordinadora] ‘compra arroz, carne, y yo preparo un arroz con carne’. Había chicos y chicas que no tienen [recursos], y en vez de estar en la calle, es mejor que estén acá, aprendiendo.” (Darío, HG, 38 años)

Coinciding with the above, we observed that some TW who did sex work remained in the community center until dinner was served and the space closed at night. That may have been their first meal of the day, and they were safeguarded in a warm place, where they could obtain condoms and lubricant, before going out to work. We do not know to what degree the unmet basic need for food among some participants influenced their attendance and participation in GA. Other classes or workshops on different topics, including make-up, hairstyling and English, were organized at the community center in response to the unmet need expressed by some GA participants to learn new skills that may help them find work. These classes were short-lived, however, since they were provided by volunteers, and GA did not have the financial resources to hire paid instructors.

## 4. Discussion

The high prevalence of HIV in MSM and TW in Peru [1] points to the need for interventions that include individual and collective empowerment strategies, in addition to community building and mobilization as relevant aspects for an effective approach to HIV prevention with the involvement of MSM and TW in developing solutions to the problems they face as vulnerable communities. The processes that we present in the results (active participation, community building and personal and community empowerment; effect on HIV prevention and treatment; integration of GM and TW and the community center as a safe space) worked in synergy as a product of the successful implementation of the core elements of GA. They generated community mobilization and positive changes in the participants related to self-empowerment, personal agency and their health. These processes align with the components identified by Lippman et al. [21] to produce community mobilization, including the development of critical consciousness about shared issues, organizational networks, leadership, collective action and social cohesion.

Our results demonstrate the feasibility and high acceptability of GA among GM and TW. Participants were highly engaged in GA and its activities that brought life to the community center. GA filled a vacuum in SJM, with the GA community center acting as a “transformative social space” [49] for support and obtaining information. The participants assumed increasing responsibility in the organization of mobilization activities, gradually developing leadership and organizational capacity. Our results also suggest a positive impact of the intervention related to HIV prevention, condom use and HIV care engagement among the participants in GA. These results need to be corroborated by greater evidence. The ongoing activities, conversations and reflections around HIV and the stigma related to HIV and towards people with HIV, together with the development of relationships of trust, allowed participants to normalize talking more openly about these topics. Several participants decided to openly share in GA that they were living with HIV. GA achieved integration and joint work between TW and GM towards a common goal: community building and community mobilization related to HIV prevention and care. This is particularly important considering that there are no precedents for projects in Peru that have achieved these levels of integration of the two populations. During the COVID-19 pandemic, the virtual activities and in-person food distribution were possible thanks to the prior development of social and support networks, the joint work between GM and TW and the community building and community mobilization achieved throughout GA implementation before March 2020.

In the context of the four community mobilization approaches [participatory action research, community-based participatory research, collaborative betterment model (CBM) and community empowerment model (CEM) described by Kim-Ju et al. [50], GA’s design and implementation fit well in a continuum between CBM and CEM. In this continuum, the researchers established the goals and guided the process of implementation of GA, but TW and GM participants, particularly through the GI, had an active role in the decision-making regarding the design, implementation and evaluation of the mobilization activities.

Depending on individual and structural factors and the level of engagement in the intervention, our results may not apply to all the TW and GM who participated in the GA activities. Nevertheless, our findings align with the results of the pilot conducted in Callao-Lima [29], which showed high acceptability of a community mobilization intervention among TW and GM. The lack of quantitative outcomes limits determining the impact of GA on HIV prevention and engagement in care. We have evidence that participants invited their friends and peers to GA and that they had informal conversations with their peers about HIV prevention and the importance of HIV treatment, but we do not have sufficient evidence about the reach and impact of the intervention among TW and GM who did not participate in GA and its actual diffusion within the community. We did not have enough financial resources to provide ongoing classes on diverse topics that could have helped participants develop life skills and perhaps eventually provide them with more economic opportunities and economic empowerment. Economic empowerment may alter the multidimensional relationship between poverty and exposure to HIV high-risk behaviors in the context of sex in exchange for money or goods, power imbalance and sexual violence [51]. Even if some activities were implemented virtually, the COVID-19 quarantine disrupted the last months of GA implementation. It did not allow us to organize a formal in-person closing of the community center and a celebration of the participants’ achievements in the project, or most importantly, consolidate the role of some participants as potential leaders and solidify external relationships with the municipality, as well as other organizations that may have been crucial to continue a process of community mobilization [49] and the potential sustainability of at least some elements of GA through the organization that the participants intended to create. However, based on anecdotal evidence and our experience observing the personal trajectory of some participants in past research projects who afterwards became community activists, researchers or educators, GA may have planted a seed and fostered a process of empowerment and personal development that may take time to flourish beyond the period of implementation and evaluation of a research project like GA [52].

Community mobilization may need to be implemented in combination with other strategies [52]. Despite its promising results, the impact of community mobilization efforts aiming to address the contexts related to HIV risk and empower people to take care of their health may vary depending on specific populations and their social conditions; how those efforts are operationalized, implemented and evaluated may also influence assessment of the impact [53]. A review of 20 HIV prevention interventions with different populations that included community mobilization and diverse measurement of outcomes showed mixed results, with some consistent and positive impact, particularly on behavioral outcomes (but more limited or inconclusive for social and biomedical outcomes) among specific populations, including sex workers and MSM. A positive behavioral impact in condom use was found in interventions in Latin America, such as the *Encontros* project with female, male and “transvestite” (a term used by the authors) sex workers in Brazil and in the Frontiers Prevention project with MSM and female sex workers in Ecuador. Positive social outcomes related to populations with a “strong collective identity”. Less conclusive results were found in interventions with youth and the general population [53].

As a political process embedded in societal relationships of power, and even with the development of agency and capacity building, sustaining the impact of community mobilization to produce social change [54] among disenfranchised and stigmatized populations in low-resource settings, such as among TW and GM of lower socioeconomic strata in Peru, may be difficult to achieve without legal, policy and structural changes [55,56]. The participants in the GI’s initiative to create their own organization after the completion of GA is laudable and reflects their intention to continue building and mobilizing as a community. This effort, however, was not fruitful in the aftermath of the COVID-19 pandemic; and it is uncertain whether that newly created organization potentially would have had the capacity or the interest to continue implementing the core elements of GA without the financial and organizational resources provided by a project such as PO+. Instead, our intention was that after conducting GA as a research project, it would be sustained and implemented on an ongoing basis by the Peruvian government and/or a community-based organization providing support and the necessary resources.

In Peru, and in Latin America in general, civil society plays an important role, and community mobilization efforts are widespread through participation in neighborhood groups, community and faith-based organizations, protests and legal and political advocacy and involvement [57]. In Peru, community-based organizations provide a variety of HIV services, including HIV testing, counseling and medical services. Utilization of these services may contribute to personal empowerment. LGBTQ organizations also provide advocacy and may foster community building, community mobilization and political participation [58]. Most of these efforts, however, are not formally evaluated. Within a political context of limited funding and deteriorating human rights in general, and particularly those of the LGBTQ community in Peru [59], our results aim to advance the study of these non-linear, multidimensional and somewhat elusive community mobilization processes in need of more specific evaluation methods [56]. Our results contribute to the literature identifying and understanding the significant community mobilization processes that may help design future and successful community mobilization interventions that address the individual, social and structural factors that increase the vulnerability to HIV of men who have sex with men and transgender women, not only in Peru but also in other Latin American countries with similar concentrated HIV epidemics [60].

## 5. Conclusions

Considering the promising results of GA, the next steps in Peru would consist of its implementation as part of government or non-governmental organization programs as a community mobilization strategy that may contribute to improving the HIV prevention and care continuum among MSM and TW. Its implementation, however, would require further evaluation. Moreover, GA could be implemented in other Latin American countries with a concentrated HIV epidemic, allowing adaptations to the needs of GM, TW and other sexual and gender minorities in each country.

## Figures and Tables

**Table 1 ijerph-22-01669-t001:** The seven theoretical and guiding principles of “Generación Actual” (GA), based on Mpowerment.

1. Multilevel approach
Consideration of the different dimensions involved in HIV prevention (individual, social/community and structural levels).
2. Social focus
Linking HIV prevention with other social needs.
3. Community building
Creation of social support networks and community contexts that promote health care.
4. Empowerment philosophy
Theories based on empowerment [32] and popular education/liberation of the oppressed [33] focused on fostering empowerment and developing personal agency.
5. Peer influence
Communication and interaction between peers are considered fundamental for social change.
6. Dissemination of the intervention in the community
Based on the Theory of the diffusion of innovations [34], change at the community level occurs through a process of informal communication and peer modeling in social networks.
7. Positive focus on self-acceptance of sexual and gender identity
Sense of affirmation and pride linked to the expression of sexuality and gender.

**Table 2 ijerph-22-01669-t002:** Seven core elements of “Generación Actual” (GA).

1. Experiential workshops or “Waves”
Periodic two/three-hour self-reflection sessions, conducted separately with 8–10 GM and TW, focused on sexual/gender identity, homophobia/transphobia, personal and community empowerment and HIV prevention and treatment. Only core GA element with a specific methodology, structure and curriculum (parallel for GM and TW). The invitation to participate in the workshops was first extended through promoting GA at spaces frequented by GM and TW in Lima South, and then through participants inviting their peers.
2. Core group (Grupo Impulsor, GI)
Comprising participants and peer coordinators who met weekly to discuss issues of interest to GM and TW, organize and implement mobilization activities and help disseminate HIV-related messages in the community.
3. Formal outreach
(a) Mobilization, education, community strengthening and mutual support activities designed and organized by the GI. (b) An outreach team in charge of visiting venues frequented by GM and TW carrying out short activities/performances to capture the attention and interest of attendees, promote GA and disseminate HIV prevention messages in a fun and entertaining way.
4. Informal outreach
Informal conversations between GA participants and their peers and friends about HIV prevention and treatment to disseminate intervention messages in the community.
5. Community center
Safe space for meeting, socializing and free expression, where most GA activities took place, in which safer sex was promoted and condoms and lubricant were available. TV, computers and access to the Internet were available for participants.
6. Ongoing promotion
Distribution of educational materials and promotion of GA through social networks and some in-person at community venues by participants in GA.
7. Peer coordinators
Paid project staff (two GM and one TW) crucial to implement GA. They facilitated community mobilization, community building and empowerment processes and promoted messages about HIV prevention and the importance of treatment. Their role was not directive or vertical, they modeled forms of behavior and the maintenance of rules of coexistence and confidentiality. They guided the joint work of GM and TW towards common goals.

**Table 3 ijerph-22-01669-t003:** Number of participants in periodic “waves” of experiential workshops.

Participants:	Gay Men		Trans Women	
	Started	Completed	Started	Completed
Wave 1	13	11	11	11
Wave 2	10	8	10	9
Wave 3	9	9	13	11
Wave 4	10	9	8	8
Wave 5	12	9	16	15
Wave 6	13	13	13	12
Wave 7	12	12	11	10
Wave 8	12	12		
Wave 9	10	10		
Wave 10	8	8		
Wave 11	11	11		
Wave 12	8	8		
TOTAL	128	120	82	76
RETENTION:	92.7%		92.7%	

Note: There were fewer waves of workshops with transgender women due to a smaller population of transgender women in the area.

**Table 4 ijerph-22-01669-t004:** Summary of activities of “Generación Actual” (GA).

Type of Activity	Number of Activities	Average Number of Participants in Activities
GI meetings	99	21
Feminine Mondays (weekly): Discussion groups that included topics such as HIV prevention and treatment, transphobia, violence, legal rights	106	10
Gay Saturdays (weekly): Discussion groups that included topics such as HIV prevention and treatment, homophobia, sexual identity, legal rights	93	17
Small activities (weekly or periodic): movie night; cooking, theater, aerobic classes, HIV support group, etc.	157	19
Medium activities (periodic): presentations, workshops, outings	23	37
Large activities (periodic): outings, gatherings, parties	13	187
Virtual activities during the COVID-19 pandemic (May–September 2020)	26	5037 (estimated reach, live participation or views)

Average number of participants in the core group (GI) and mobilization activities, and average number of small, medium, large and virtual mobilization activities.

## Data Availability

The original contributions presented in this study are included in the article. The data collected and analyzed for this study are not publicly available to protect participant confidentiality. Further inquiries can be directed to the corresponding author.

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
