# Peer review of "A Qualitative Assessment of “Generacion Actual”: An HIV Community Mobilization Intervention Among Gay Men and Transgender Women in Lima, Peru"

_ijerph, 2025, doi:10.3390/ijerph22111669_

Round 1
Reviewer 1 Report
Comments and Suggestions for Authors
The study presented in this manuscript is a qualitative analysis of in-depth interviews with 24 gay men (GM) and transgender women (TW) who participated in Generación Actual (GA), an HIV prevention intervention for gay men GM and TW in Lima South, Peru. Interviews examined perceptions of and experiences with participation in GA and revealed four significant processes that facilitated community mobilization: (1) active participation, community building and personal and community empowerment; (2) effect of GA on HIV prevention and treatment; (3) integration of GM and TW in GA and (4) the GA community center as a safe socialization space. Overall, the manuscript is well-written, but requires extensive revisions to improve clarity, particularly on the study purpose, methods, and presentation of findings.
Abstract
- The sample size should be specified.
Introduction
- Consider replacing outdated terminology (e.g., “HIV/AIDS,” “HIV patients”) with more precise, person-first language, as per UNAIDS terminology guidelines (https://www.unaids.org/sites/default/files/media_asset/2024-terminology-guidelines_en.pdf).
- There are a few sentences that require a citation (e.g., “Condom use among MSM and TW in Peru is inconsistent and is usually discontinued when in a relationship.”)
- It may be helpful to direct readers to Tables 1 and 2 when GM theoretical and guiding principles and core elements are first mentioned (in the sentence beginning on line 110).
- Consider replacing “significative” with “significant.”
- “The purpose of this qualitative analysis is to examine four significative community mobilization processes...” If these processes are findings that were identified through the present study, they should not be included in the study purpose. Relevant text throughout the manuscript should be revised accordingly.
- Consider numbering the processes listed in the following sentence, as it is unclear what the four distinct processes are: “The purpose of this qualitative analysis is to examine four significative community mobilization processes that occurred throughout the implementation of GA. These processes (active participation, community building, personal and community empowerment; effect on HIV prevention and treatment; integration of gay men and transgender women and the community center as a safe space)…”
Materials and Methods
- Additional details are needed throughout this section. For example:
- Specify the sample size.
- Provide more information on purposive sampling: “We asked the coordinators in charge of organizing GA to identify TW and GM of different ages participating in different GA activities to be part of the interviews.”
- How were participants recruited?
- Is the interview guide available for review?
Results
- Much of the information provided about GA throughout this section is better suited for the Introduction or the Materials and Methods section (or another manuscript). For example, many of the details in subsection 3.1 and quantitative results (i.e., Tables 3 and 4) are not appropriate for this section, which should focus on presenting findings from the qualitative analysis of in-depth interviews. I strongly suggest that the authors reorganize the manuscript to streamline and clearly present details about GA to provide context in the Introduction or the Materials and Methods section and then focus the remainder of the manuscript on the present study.
Discussion
- Currently, the authors allude to limitations but they should be more explicitly discussed.
Conclusion
- Given the lack of quantitative outcome assessment to determine the impact of GA, the Conclusion is likely overstated as written. The Conclusion should be more closely aligned with the findings of the present study.
Author Response
REVIEWER 1
We greatly appreciate the reviewer’s constructive comments and the opportunity to submit revisions to the manuscript originally titled “Generacion Actual": An HIV Community Mobilization Intervention among Gay Men and Transgender Women in Lima, Peru, ” now renamed as “A Qualitative Assessment of "Generacion Actual": An HIV Community Mobilization Intervention among Gay Men and Transgender Women in Lima, Peru.”
Below you will find point-by-point responses addressing the reviewer’s comments. We have also edited and made changes throughout the manuscript for clarity. Based on a comment from the second reviewer, we have changed the title of the paper to: “A Qualitative Assessment of "Generacion Actual": An HIV Community Mobilization Intervention among Gay Men and Transgender Women in Lima, Peru.”
Open Review
(x) I would not like to sign my review report
( ) I would like to sign my review report
Quality of English Language
( ) The English could be improved to more clearly express the research.
(x) The English is fine and does not require any improvement.
|
Yes |
Can be improved |
Must be improved |
Not applicable |
|
|
Does the introduction provide sufficient background and include all relevant references? |
( ) |
( ) |
(x) |
( ) |
|
Is the research design appropriate? |
(x) |
( ) |
( ) |
( ) |
|
Are the methods adequately described? |
( ) |
( ) |
(x) |
( ) |
|
Are the results clearly presented? |
( ) |
( ) |
(x) |
( ) |
|
Are the conclusions supported by the results? |
( ) |
(x) |
( ) |
( ) |
|
Are all figures and tables clear and well-presented? |
(x) |
( ) |
( ) |
( ) |
Comments and Suggestions for Authors
The study presented in this manuscript is a qualitative analysis of in-depth interviews with 24 gay men (GM) and transgender women (TW) who participated in Generación Actual (GA), an HIV prevention intervention for gay men GM and TW in Lima South, Peru. Interviews examined perceptions of and experiences with participation in GA and revealed four significant processes that facilitated community mobilization: (1) active participation, community building and personal and community empowerment; (2) effect of GA on HIV prevention and treatment; (3) integration of GM and TW in GA and (4) the GA community center as a safe socialization space. Overall, the manuscript is well-written, but requires extensive revisions to improve clarity, particularly on the study purpose, methods, and presentation of findings.
Abstract
-The sample size should be specified.
Response: We have added the sample size in the abstract.
Introduction
-Consider replacing outdated terminology (e.g., “HIV/AIDS,” “HIV patients”) with more precise, person-first language, as per UNAIDS terminology guidelines (https://www.unaids.org/sites/default/files/media_asset/2024-terminology-guidelines_en.pdf).
Response: We thank the reviewer for this comment. We have consulted the UNAIDS terminology and replaced a few terms: HIV/AIDS cases with HIV cases, patients with HIV with patients and, in one instance, safe sex with safer sex. We did not find in the manuscript any other terms that needed revising according to UNAIDS guidelines. We still use the term “intervention” as appropriate, according to UNAIDS, for health-related interventions.
-There are a few sentences that require a citation (e.g., “Condom use among MSM and TW in Peru is inconsistent and is usually discontinued when in a relationship.”)
Response: We have added a citation to the “Condom use..” sentence, as well as to other sentences.
-It may be helpful to direct readers to Tables 1 and 2 when GM theoretical and guiding principles and core elements are first mentioned (in the sentence beginning on line 110).
Response: There is already a sentence in the text directing readers to Tables 1 and 2.
-Consider replacing “significative” with “significant.”
Response: We thank the reviewer for this comment and had replaced significative with significant throughout. We had, in fact, during writing, gone back and forth many times whether to use significant or significative. Since significative, while a valid word expressing symbolism may be considered antiquated or incorrect, we have replaced it with significant, as a more standard and widely used and understood word.
-“The purpose of this qualitative analysis is to examine four significative community mobilization processes...” If these processes are findings that were identified through the present study, they should not be included in the study purpose. Relevant text throughout the manuscript should be revised accordingly.
-Response: The four community mobilization processes are findings. We had included them in the study purpose in an attempt to address/interpret the following journal guideline: “Briefly mention the main aim of the work and highlight the main conclusions.” Since we have noted that other qualitative articles in IJERPH do not include this, we have edited that sentence as a more general statement to reconcile it with the reviewer’s comment. It may be to the editor’s discretion to include or modify that sentence in the Introduction section.
-Consider numbering the processes listed in the following sentence, as it is unclear what the four -distinct processes are: “The purpose of this qualitative analysis is to examine four significative community mobilization processes that occurred throughout the implementation of GA. These processes (active participation, community building, personal and community empowerment; effect on HIV prevention and treatment; integration of gay men and transgender women and the community center as a safe space)…”
Response: For clarity, we have edited the sentence above and numbered the four processes as suggested by the reviewer.
Materials and Methods
-Additional details are needed throughout this section. For example:
- Specify the sample size. Response: We have added the sample size in the methods. For clarity and as a reminder to the reader but at the risk of duplication, have also kept it in the results, as was done in other articles published in IJERPH, also because of context, since when stating the size of the sample, we also refer to the ages and occupation of the participants, which is data part of the results.
- Provide more information on purposive sampling: “We asked the coordinators in charge of organizing GA to identify TW and GM of different ages participating in different GA activities to be part of the interviews.”
- How were participants recruited? Response: We have added more information on purposive sampling and on how the participants were recruited.
- Is the interview guide available for review? Response: We have included the interview guide for review, at the discretion of the editor whether to include it as supplemental material.
Results
-Much of the information provided about GA throughout this section is better suited for the Introduction or the Materials and Methods section (or another manuscript). For example, many of the details in subsection 3.1 and quantitative results (i.e., Tables 3 and 4) are not appropriate for this section, which should focus on presenting findings from the qualitative analysis of in-depth interviews. I strongly suggest that the authors reorganize the manuscript to streamline and clearly present details about GA to provide context in the Introduction or the Materials and Methods section and then focus the remainder of the manuscript on the present study.
Response: We appreciate the reviewers’ comment. We have moved some of the information about GA in section 3.1 to the Introduction but have kept some information about GA in the results that helps understand and provides context to GA implementation. Information on GA in the Introduction refers to the design of GA, including its guiding principles and core elements. Consistent with qualitative methods, in this article we are telling a story. The specific information on GA activities and the reference to its core elements in the results are essential as we tell the story of GA, and how its implementation resulted in community mobilization processes. This information serves as a guide and a reminder to the reader and provides context to the findings to understand how GA and its core elements were implemented.
Similarly, the quantitative tables 3 and 4 with number of participants in the experiential workshops and mobilization activities in section 3.1 are part of the results. As included in the Methods, they inform, complement and provide context to the findings from the qualitative interviews included in the same section. We followed the journal’s guidelines inserting the tables where they are first cited.
Discussion
-Currently, the authors allude to limitations but they should be more explicitly discussed.
Response: We have elaborated on the limitations.
Conclusion
-Given the lack of quantitative outcome assessment to determine the impact of GA, the Conclusion is likely overstated as written. The Conclusion should be more closely aligned with the findings of the present study.
Response: We respectfully disagree with the reviewers’ comment. We do not consider that the conclusion is overstated as written since it refers to “the promising results of GA….as a community mobilization strategy that may contribute to improving the HIV prevention and care continuum among MSM and TW.” However, we have added in that sentence that future implementation of GA would need further evaluation. Additionally, we have added in the limitations, although not as a limitation of this article per se that focuses on community mobilization, that the lack of quantitative outcomes limits determining the impact of GA on HIV prevention and engagement in care. The discussion also states that the results need to be corroborated with greater evidence.
Reviewer 2 Report
Comments and Suggestions for Authors
The authors submitted an interesting study however the title of the study insufficiently reflect the type of study reported. The abstract insufficiently represents the results presented in the study. Proper research reporting checklist should be included with the submission.
It is not clear how the US programme was adapted to Peruvian context and it should be either based on existing publication or explain briefly how it was adapted.
The authors insufficiently described what it is research gap they are addressing and how it was identified (was systematic review/metasynthesis performed to prepare for this study? were any similar studies published so far?) The authors have not explained what this study adds to the current literature.
The authors should report their study according to the proper reporting checklists, i.e. SROR and COREQ. Right now several elements listed in the reporting checklists are not explicitly reported.
The authors summarised theoretical concepts for the programme and its core elements. The main question of the research is generally stated however more details regarding specific questions explored by the study should be provided (item 4 of SROR).
Since the authors specified their study as qualitative analysis as per SROR checklist the authors should specify their qualitative approach for the study and research paradigm (item 5 SROR).
The authors should provide specific information regarding researchers characteristics and
Reflexivity (item 6 SROR), context (item 7), sampling strategy (item 8).
The authors provided information regarding ethical issues (item 9) and information about data collection (item 10), but insufficient information regarding modifications of procedures and insufficient information regarding data collection instruments and technologies (item 11). The authors provided information about the number of participants and their basic characteristics (item 12), but insufficient information regarding other materials used in the analyses.
The authors provided information about transcribing the interviews (item13) but insufficient information regarding other methods of data processing (see item 13 SROR), paradigm or approach for data analysis (item 14) and techniques to enhance trustworthiness and credibility of data analysis (item 15).
The results contain a mixture of process implementation data and qualitative data analysis, information about the interventions should be moved to the methods section describing the context of the study. In the results section the authors provided citations but insufficiently summarised main findings (e.g. interpretations, inferences, and themes) (item 16, 17)
The discussion should be improved and should include comparison with other studies in the field, previous challenges addressed by this study, unique contributions of this study, limitations (item 18,19). The authors provided clear and specific COI and funding information (item 20,21).
The authors should also supplement this information required by SROR checklist by the information more specific for the interviews based on COREQ checklist, such as research team and reflexitivity (domain 1 item 1-8), study design (domain 2 items 9-23), analysis and findings (domain 3 items 24-32).
Additional information can be provided in the supplements
Comments on the Quality of English LanguageEnglish can be improved.
Author Response
REVIEWER 2
We greatly appreciate the reviewer’s comments and the opportunity to submit revisions to the manuscript originally titled ““Generacion Actual": An HIV Community Mobilization Intervention among Gay Men and Transgender Women in Lima, Peru, ” which, following the reviewer’s comment we have now renamed as “A Qualitative Assessment of "Generacion Actual": An HIV Community Mobilization Intervention among Gay Men and Transgender Women in Lima, Peru.” Below you will find point-by-point responses addressing the reviewer’s comments. We have also edited and made changes throughout the manuscript for clarity.
Open Review
(x) I would not like to sign my review report
( ) I would like to sign my review report
Quality of English Language
(x) The English could be improved to more clearly express the research.
( ) The English is fine and does not require any improvement.
|
Yes |
Can be improved |
Must be improved |
Not applicable |
|
|
Does the introduction provide sufficient background and include all relevant references? |
(x) |
( ) |
( ) |
( ) |
|
Is the research design appropriate? |
( ) |
(x) |
( ) |
( ) |
|
Are the methods adequately described? |
( ) |
( ) |
(x) |
( ) |
|
Are the results clearly presented? |
( ) |
( ) |
(x) |
( ) |
|
Are the conclusions supported by the results? |
( ) |
(x) |
( ) |
( ) |
|
Are all figures and tables clear and well-presented? |
(x) |
( ) |
( ) |
( ) |
Comments and Suggestions for Authors
-The authors submitted an interesting study however the title of the study insufficiently reflect the type of study reported. The abstract insufficiently represents the results presented in the study. Proper research reporting checklist should be included with the submission.
Response: We have edited the title as stated above to include this is a qualitative assessment.
We briefly summarize the results in the abstract based on the IJERPH maximum of 200 words in an abstract. While we have included items included in qualitative reporting checklists, IJERPH does not require a reporting checklist in its publication requirements.
-It is not clear how the US programme was adapted to Peruvian context and it should be either based on existing publication or explain briefly how it was adapted.
Response: We thank the reviewer for this observation and have added some examples of how GA, based on a US intervention, was adapted to the Peruvian context.
-The authors insufficiently described what it is research gap they are addressing and how it was identified (was systematic review/metasynthesis performed to prepare for this study? were any similar studies published so far?) The authors have not explained what this study adds to the current literature.
Response: We have further elaborated in the Introduction the rationale for Projecto Orgullo + addressing a research gap in Peru and the lack of community mobilization strategies to improve the continuum of HIV prevention in care among men who have sex with men and transgender women. We have added in the Introduction that PO+ was a collaboration between HIV researchers at the University of California and HIV researchers in Peru, researchers who are familiar with research gaps and the continuum of prevention in care in Peru. Various references, including the ones below, refer to the continuum of prevention and care in Peru and the limits of Peru’s HIV prevention strategy with men who have sex with men and transgender women. Some of the authors in the referenced articles are leading HIV researchers in Peru who have been involved in our project.
Continuo de la atención de personas que viven con VIH y brechas para el logro de las metas 90-90-90 en Perú. Garcia-Fernandez L, Novoa R, Huaman B, Benites C. Rev Peru Med Exp Salud Publica. 2018;35(3): 491-6. doi:10.17843/rpmesp.2018.353.3853
Chow, J.Y., Konda, K.A., Borquez, A., Caballero, P., Silva-Santisteban, A., Klausner, J.D, & Cáceres, C.F. Peru's HIV care continuum among men who have sex with men and transgender women: opportunities to optimize treatment and prevention. Int J STD AIDS. 2016 Oct;27(12):1039-1048. doi: 10.1177/0956462416645727
Response: We have further elaborated in the discussion what this article adds to the current literature.
-The authors should report their study according to the proper reporting checklists, i.e. SROR and COREQ. Right now several elements listed in the reporting checklists are not explicitly reported.
Response: We have added multiple items included in the SROR and COREQ checklists to the reporting of the metehods. Since not all qualitative research is the same, we interpret SROR and COREQ as guidelines to consider. See Reference: Reporting guidelines for qualitative research: a values-based approach. https://www.tandfonline.com/doi/full/10.1080/14780887.2024.2382244. We have not included the SROR or COREQ checklists since, like other journals, they are not part of IJERPH submission guidelines.
-The authors summarised theoretical concepts for the programme and its core elements. The main question of the research is generally stated however more details regarding specific questions explored by the study should be provided (item 4 of SROR).
Response: We have further clarified in the data collection section of the methods the domains included in the interview guide. HIV-related issues included in the experiential workshops are also mentioned in the GA section of the Introduction.
-Since the authors specified their study as qualitative analysis as per SROR checklist the authors should specify their qualitative approach for the study and research paradigm (item 5 SROR).
Response: We now have made explicit the research paradigm informing our study. We state in the analysis section of the methods that the analysis process was iterative, using an inductive and deductive approach based on thematic analysis. We have elaborated on the content of thematic analysis.
-Your assumptions about the researcher's values and their role in the research process. This is crucial for acknowledging researcher bias in interpreting findings.
Response: We have added that the interviewers and analysts are native Spanish speakers with experience conducting qualitative research. Authors are researchers with a long trajectory and experienced conducting evaluation studies and HIV research with GM and TW and other vulnerable populations in Latin America. We consider that this statement is most relevant to the specifics of this article and helps clarify how we are positioned regarding the research and the participants. Our results are data-based, and we bring the voices of the participants as included in the quotes both in English and Spanish. Data analysis, writing and editing the article as a team helped to ensure reflexivity and avoid any potential bias or pre-conceptions in the interpretation of the findings. Because of the complicated political climate in Peru and the U.S, we do include any other personal information about the authors.
- Introduction:
Clearly identify the paradigm you're using and provide a strong justification for your choice, linking it to your research questions and goals.
Response: We have identified a research paradigm applied in qualitative research.
-The authors should provide specific information regarding researchers characteristics and
Reflexivity (item 6 SROR), context (item 7), sampling strategy (item 8).
Response: We have added items related to the SROR checklist that refer to the comments above.
-The authors provided information regarding ethical issues (item 9) and information about data collection (item 10), but insufficient information regarding modifications of procedures and insufficient information regarding data collection instruments and technologies (item 11). The authors provided information about the number of participants and their basic characteristics (item 12), but insufficient information regarding other materials used in the analyses.
Response: We have added items related to the SROR checklist that refer to the comments above. There were no modifications in procedures during data collection.
-The authors provided information about transcribing the interviews (item13) but insufficient information regarding other methods of data processing (see item 13 SROR), paradigm or approach for data analysis (item 14) and techniques to enhance trustworthiness and credibility of data analysis (item 15).
Response: We have added multiple items related to the SROR checklist that refer t the comments above.
-The results contain a mixture of process implementation data and qualitative data analysis, information about the interventions should be moved to the methods section describing the context of the study. In the results section the authors provided citations but insufficiently summarised main findings (e.g. interpretations, inferences, and themes) (item 16, 17)
Response: We appreciate the reviewers’ comment. We have moved some of the information about GA in section 3.1 to the Introduction, but have kept some information about GA in the results that helps understand and provides context to GA implementation. Information on GA in the Introduction refers to the design of GA, including its guiding principles and core elements. Consistent with qualitative methods, in this article we are telling a story. The specific information on GA activities and the reference to its core elements in the results are essential as we tell the story of GA, and how its implementation resulted in community mobilization processes. This information serves as a guide and a reminder to the reader and provides context to the findings to understand how GA and its core elements were implemented.
The results mainly focus on qualitative findings from the interviews we conducted informed by observations and some quantitative data. As we state in the methods, all data presented in the results were part of a process evaluation. The quantitative tables 3 and 4 with number of participants in the experiential workshops and mobilization activities in section 3.1 are part of the results. They inform, complement and provide context to the findings from the qualitative interviews included in the same section.
We respectfully disagree with the reviewer and believe that we have sufficiently summarized the main findings, which are presented according to the four themes in the results section. The findings are data-based and we cannot over-interpret or infer what is not in the data.
-The discussion should be improved and should include comparison with other studies in the field, previous challenges addressed by this study, unique contributions of this study, limitations (item 18,19). The authors provided clear and specific COI and funding information (item 20,21).
Response: We have reorganized and added text in the discussion section in response to the reviewer’s comment above.
-The authors should also supplement this information required by SROR checklist by the information more specific for the interviews based on COREQ checklist, such as research team and reflexitivity (domain 1 item 1-8), study design (domain 2 items 9-23), analysis and findings (domain 3 items 24-32).
Response: We have added items related to the SROR and COREQ checklists that refer to the comments above.
Additional information can be provided in the supplements
Comments on the Quality of English Language
English can be improved.
Round 2
Reviewer 1 Report
Comments and Suggestions for Authors
I thank the authors for addressing previous comments. The manuscript has been substantially strengthened through revision.